# Medically Attended Outpatient Parainfluenza Virus Infections in Young Children from a Single Site in Machala, Ecuador

**DOI:** 10.3390/ijerph22060821

**Published:** 2025-05-23

**Authors:** Manika Suryadevara, Dongliang Wang, Freddy Pizarro Fajardo, Jorge Luis Carrillo Aponte, Froilan Heras, Cinthya Cueva Aponte, Irene Torres, Joseph Domachowske

**Affiliations:** 1Department of Pediatrics, State University of New York Upstate Medical University, Syracuse, NY 13066, USA; domachoj@upstate.edu; 2Department of Public Health and Preventative Medicine, State University of New York Upstate Medical University, Syracuse, NY 13066, USA; wangd@upstate.edu; 3Hospital General del Norte de Guayaquil IESS Los Ceibos, Guayaquil 090101, Ecuador; fredduos@hotmail.com; 4Research Center at Hospital Teofilo Davila, State University of New York Upstate Medical University, Machala 070102, Ecuador; jorge.carrillo@ucuenca.edu.ec (J.L.C.A.); herasfroilan@gmail.com (F.H.); cin_ka10@hotmail.com (C.C.A.); 5Fundacion Octaedro, Quito 170150, Ecuador; irene.torres@octaedro.edu.ec

**Keywords:** parainfluenza virus, tropics, pediatrics, respiratory infection, Ecuador

## Abstract

Parainfluenza virus (PIV) infections contribute to the overall childhood morbidity from acute respiratory illness, yet virus-specific epidemiologic data are lacking across many regions globally. Here, we describe the clinical manifestations, seasonality, and meteorologic associations with PIV infections in Ecuadorian children. Between July 2018 and July 2023, we documented demographic and clinical information from children younger than 5 years seen in a single public health clinic with signs and symptoms consistent with an acute respiratory infection. Nasopharyngeal swabs collected at study enrollment underwent multiplex polymerase chain reaction-based diagnostic testing (Biofire FilmArray v. 1.7™). Regional meteorological data from the same period were provided by Ecuador’s Instituto Nacional de Meteorologia e Hidrologia. Parainfluenza viruses were detected in 9% of the 1251 enrolled subjects. PIVs were most frequently detected between March and July, with no change in seasonality following SARS-CoV-2 pandemic onset. Clinical manifestations of PIV infections included non-specific upper respiratory illness (82%), laryngotracheitis (3%), and bronchiolitis (11%). Events of PIV detection were negatively associated with ambient temperature and rainfall. Our findings highlight the contribution that PIVs play in the morbidity associated with pediatric medically attended outpatient respiratory tract infection and provide new insights into the seasonal epidemiology of PIV infections in coastal Ecuador.

## 1. Introduction

Parainfluenza viruses (PIVs) are among the leading causes of lower respiratory tract infections globally, second only to respiratory syncytial virus (RSV), with seasonality of infection varying by virus subtype and geographic region [1,2]. Associated with both upper and lower respiratory tract disease, PIV infections can manifest as otitis media, pharyngitis, laryngotracheobronchitis, bronchiolitis, and/or pneumonia, leading to a large number of medically attended visits and hospitalizations and significantly adding to the economic burden of families, communities, health systems, and national income [1,2].

In the United States, it is estimated that PIVs account for approximately 17% of outpatient viral respiratory disease and 7% of hospitalizations for febrile or respiratory illness in children younger than 5 years. Such infections lead to healthcare costs exceeding 200 million USD annually [2]. Given limited diagnostic testing and reporting of PIV infections throughout much of the world, there are very limited data regarding PIV epidemiology, seasonality, and disease burden globally [3]. A better understanding of regional epidemiology is necessary to optimize the implementation of disease prevention measures, improve the evaluation of resources required for the delivery of needed health care, address and assess healthcare utilization, and promote judicious and appropriate use of antibiotics in the setting of acute respiratory tract illness [3].

Machala is a coastal equatorial city located in the upper middle-income country of Ecuador. Despite existing reports that lower respiratory tract infections are a leading cause of communicable disease and mortality in Ecuador, detailed respiratory virus-specific epidemiologic data from this region are lacking [4,5]. In this study, we describe the epidemiology and clinical manifestations of PIV infections and assess possible meteorological influences on these infections among young children living in coastal Ecuador for periods of time before and after the onset of the SARS-CoV-2 pandemic.

## 2. Materials and Methods

This study is part of a 5-year cross-sectional surveillance parent study of the etiology and clinical presentation of medically attended outpatient respiratory tract infections among Ecuadorian children less than 5 years of age. The methods for study recruitment and procedures were previously described [6,7]. Subjects were recruited from a single public outpatient clinic in Machala, Ecuador between July 2018 and July 2023. Study enrollment was temporarily paused between March and August 2020 due to restrictions imposed in response to the SARS-CoV-2 pandemic. Data obtained prior to the pause in enrollment are referred to as ‘before the onset of the SARS-CoV-2 pandemic’, while data obtained after enrollment re-started are referred to as ‘after the onset of the SARS-CoV-2 pandemic’. This project is approved by the State University of New York Upstate Medical University Institutional Review Board (IRB number 1102402) and by the Ministry of Health of Ecuador.

### 2.1. Subject Enrollment

Children younger than 5 years age who were evaluated at a study-designated public outpatient clinic for signs and symptoms consistent with an acute respiratory tract infection (ARTI) were eligible for enrollment. An ARTI was defined by the presence or parental report of two or more of the following symptoms for fewer than 8 days: temperature ≥ 38 °C, nasal congestion or discharge, cough, tachypnea, wheezing, rales, hypoxia, or apnea. Subjects were excluded from enrollment if they were in foster care, had parents who were unable or unwilling to provide consent, or were hospitalized or treated with antibiotics within the last 30 days.

The study team obtained informed consent, then collected and recorded demographic (age, gender, and visit date) and clinical data (symptoms, symptom duration, primary medical diagnosis code, treatment plans, and influenza and pertussis vaccination status). For study purposes, upper respiratory tract infections (URTI) included primary medical diagnoses of nasopharyngitis and laryngotracheitis. Lower respiratory tract infections (LRTI) included diagnoses of bronchitis, bronchiolitis, and pneumonia.

A nasopharyngeal sample was obtained, placed in 3 mL of universal transport, processed and analyzed for the presence of respiratory pathogen nucleic acids using the multiplex PCR-based BioFire Film Array™ Respiratory Panel v1.7 (BioFire Diagnostics LLC, Salt Lake City, UT, USA). This multiplex panel is designed to detect adenoviruses, coronaviruses (HKU1, NL63, 229E, and OC43), human metapneumovirus (hMPV), rhinovirus/enterovirus, influenza viruses (A-H1N1, A-H3N1, B), parainfluenza viruses (PIV) (types 1–4), respiratory syncytial virus, *Bordetella pertussis*, *Chlamydophila pneumoniae*, and *Mycoplasma pneumoniae*.

### 2.2. Meteorologic Data Collection

Data downloaded from weather towers maintained by Ecuador’s Instituto Nacional de Meteorologia e Hidrologia provided the meteorologic data including temperature (°C), relative humidity (%), barometric pressure (W/m2), and precipitation (mm) for this analysis. The weather tower instruments were calibrated every 3 months and immediately following any necessary repairs. Meteorologic data were collected every minute, continuously, and then summarized as daily, weekly, and/or monthly averages. Means, medians, and ranges were calculated for each week. From the measured data, dew points were calculated using the following formula:
B = (ln (RH/100) + ((17.27 × T)/(237.3 + T)))/17.27
    D = (237.3 × B)/(1 − B)
where T = air temperature in °C; RH = relative humidity in percent (%); B = intermediate value (no units) and D = dewpoint in °C.

### 2.3. Statistical Analysis

Descriptive statistics were used. Continuous data were compared by *t*-test and categorical data were compared using the Fisher exact or chi-square test, as appropriate. Significance was set a priori to a *p* value of <0.05. Receiver operating characteristics (ROC) analyses were performed to evaluate the performance of the potential classification factors/models by calculating the sensitivities (true positive rates) and specificities (true negative rates) at each classification threshold. The area under the ROC curve represents an overall measure of the classification accuracy and a bootstrap-based test was performed to assess whether the AUC was different from 0.5, the AUC of a noninformative classification model. Furthermore, the optimal threshold was chosen to maximize the sum of sensitivity and specificity. All factors with significant differences between the two groups were included in multiple logistic regression to derive a composite score for classification, as a weighted average of individual factors. The differences between different logistic models were tested by Rao’s score test.

## 3. Results

### 3.1. All Enrolled Subjects

There was a total of 1251 subjects, with a mean age of 19 months, enrolled in this study. Of the enrolled subjects, 117 (9%) samples tested positive for the detection of a parainfluenza virus (Table 1). Of these, 12% were PIV type 1, 7% were PIV type 2, 61% were PIV type 3, and 21% were PIV type 4. Subjects whose nasopharyngeal samples were positive for the detection of PIV were more likely than those whose samples did not detect PIV to report wheezing (20/117 (17%) vs. 73/1134 (6%), *p* < 0.0001). There were no other statistical associations between demographic and clinical data and PIV detection, specifically no difference in PIV detection by age, gender, diagnosis, and receipt of antibiotics (Table 1).

### 3.2. Subjects Whose Nasopharyngeal Samples Were Positive for the Detection of PIV

The nasopharyngeal samples from 117 subjects were positive for the detection of a PIV. The mean age of subjects with documented PIV infection was 18 months (range of 1 to 60 months). 57 (49%) of the subjects who tested positive for PIV were younger than 1 year of age, while 60 (51%) were between the ages of 1 and 5 years. Subjects between the ages of 1 and 5 years were more likely than those younger than 1 year of age to report fever with their respiratory illness (54/60 (90%) vs. 41/57 (72%), *p* = 0.01) and were more likely to have been vaccinated against influenza virus if age eligible (55/60 (92%) vs. 8/31 (26%), *p* < 0.0001) and pertussis (58/60 (97%) vs. 48/57 (84%), *p* 0.03). There were no other significant differences between subject age, symptoms, symptom duration, or primary medical diagnosis.

Of the 117 subjects whose nasopharyngeal samples were positive for the detection of a PIV, 14 (12%) detected PIV type 1, 8 (7%) detected PIV type 2, 71 (61%) detected PIV type 3, and 25 (21%) detected PIV type 4 (Table 2). PIV 1 and PIV 4 were more likely to be associated with reports of wheezing with acute respiratory illness and a diagnosis of lower respiratory tract infection compared to PIV 2 and PIV 3 (*p* < 0.05) (Table 2). There were no other statistical associations between PIV type, age, or gender.

Of the 117 parainfluenza virus positive samples, 29 (25%) also detected at least one other pathogen: 18 (62%) rhinovirus/enterovirus, 5 (17%) RSV, 3 (10%) influenza virus, and 1 (3%) each of human coronavirus, adenovirus, and atypical bacteria. While most of these samples detected 2 co-pathogens, 2 samples detected 3 pathogens (PIV 3 with rhinovirus/enterovirus and RSV; PIV 1 with human metapneumovirus and adenovirus). There were no significant differences between subjects with pathogen co-detection or just detection of a parainfluenza virus with regards to age (mean 17.2 months vs. 17.7 months, *p* = 0.9), fevers (24/29 (83%) vs. 71/88 (81%), *p* = 0.8), nasal congestion (28/29 (97%) vs. 82/88 (93%), *p* = 0.7), cough (19/29 (66%) vs. 51/88 (58%), *p* = 0.5), wheeze (8/29 (28%) vs. 12/88 (14%), *p* = 0.08), or symptom duration (mean 3.6 days vs. 3.2 days, *p* = 0.3).

Among the 117 nasopharyngeal samples that tested positive for the detection of PIV, 57 (49%) were collected prior to the onset of the SARS-CoV-2 pandemic (Table 3). Subjects testing positive for PIV who were enrolled after pandemic onset were more likely to report fever (58 (97%) vs. 37 (65%), *p* < 0.00001) and less likely to report cough (25 (42%) vs. 45 (79%), *p* < 0.0001) with their illness. Similarly, those enrolled after pandemic onset were more likely to be diagnosed with an upper respiratory tract infection (59 (98%) vs. 37 (65%), *p* < 0.001) and less likely to be diagnosed with a lower respiratory tract infection (2 (3%) vs. 14 (25%), *p* < 0.001).

### 3.3. PIV Seasonality

Of the 1251 nasopharyngeal samples obtained over a period of 221 weeks, the 117 samples that detected PIV were collected over 72 (33%) different calendar weeks (Figure 1). Most of the PIV activity occurred between March and July. There were no significant changes in general or type-specific PIV seasonality following pandemic onset.

**Figure 1 ijerph-22-00821-f001:**
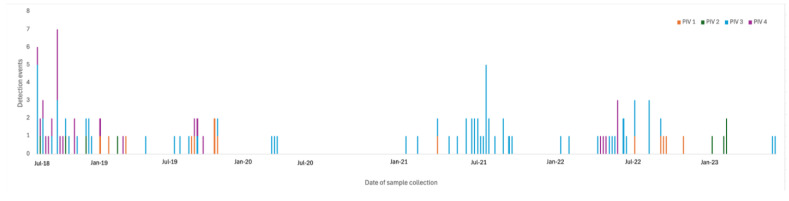
Shown is the study timeline (*x*-axis) from July 2018 through July 2023 indicating parainfluenza virus (PIV) detection events by type (*y*-axis).

### 3.4. Meteorologic Associations

Over the course of the study period, there was evidence of seasonal variation in outdoor air temperature with the highest temperatures occurring during March (mean 28 °C, maximum 34 °C) and the lowest occurring during August (mean 23 °C, minimum 20 °C). Mean relative humidity was found to increase from 71% in January (range 8% to 100%) gradually throughout the year to a high of 85% in November (range 10% to 100%), while mean barometric pressure remained between 1010 W/m^2^ and 1012 W/m^2^.

PIV detection events were associated with periods of lower mean outdoor air temperature (with a temperature threshold of 25 °C), larger humidity range, and higher mean barometric pressure (Table 4). After both univariate and multivariate logistic regression analysis was performed, only mean temperature and humidity range remained associated with PIV detection (Table 5). PIV detection was also associated with rainfall less than 1.15 mm per week, with a 5-week lag (AUC = 0.68; 95% CI 0.60–0.76, *p* < 0.001). PIV 3, specifically, was detected at a higher frequency during the dry months of June through August than the rainy months of January through April (43/314 (14%) vs. 11/399 (3%), *p* < 0.0001).

## 4. Discussion

Over the 5-year study period, we found that PIV 3 was the most frequently detected of the 4 parainfluenza virus types. PIV 3 was more likely than the other virus subtypes to manifest as an upper respiratory infection and was more frequently detected during the dry months than the rainy months. Overall, PIV detection was associated with lower mean outdoor air temperature and larger daily humidity range.

We detected PIV in 9% of subjects seeking medical attention for an acute respiratory infection, a rate higher than what has been previously reported [8,9]. Azziz-Baumgartner found that 6% of children ages 1–4 years with a medically attended acute respiratory tract infection in northwest Ecuador tested positive for parainfluenza virus [8]. A respiratory pathogen surveillance study found that 1.5% of tested samples were positive for parainfluenza virus in Ecuador’s coastal region [9]. Differences in diagnostic testing likely account for the increase in PIV detection in our study with the use of multiplex-PCR assays, which are more sensitive for virus detection than the direct immunofluorescence assays used in the prior studies. Improved diagnostic testing methods lead to a truer understanding of virus-specific epidemiology, data which can be used to guide both prevention and treatment measures. One study published in 2017 found that 57/406 (14%) of children under 5 years of age admitted to the hospital in Quito, Ecuador, for severe pneumonia tested positive for a parainfluenza virus [10]. It is important to note, however, the difference in the study population as Jonnalagadda’s study focused on hospitalized children and our study on children seeking ambulatory medical care.

Participants with nasopharyngeal samples positive for the detection of PIV were more likely to report wheeze than those whose samples did not detect this virus. In addition, 11% of participants with PIV detected in their nasopharyngeal specimen were diagnosed with bronchiolitis. While RSV is the most common cause of viral bronchiolitis, these data support the previously described contribution of PIV to the disease burden of bronchiolitis in young children [11].

Individuals with nasopharyngeal samples positive for the detection of PIV who were enrolled prior to the onset of the SARS-CoV-2 pandemic were more likely to report cough and more likely to be diagnosed with a lower respiratory tract infection, while those enrolled after the pandemic onset were more likely to report fever and more likely to be diagnosed with an upper respiratory tract infection. Similarly, we previously reported in the parent study, that participants whose nasopharyngeal samples detected RSV prior to the onset of the pandemic were more likely to report cough or wheeze and be diagnosed with a lower respiratory tract infection, while those infected with RSV after pandemic onset were more likely to report fever and be diagnosed with an upper respiratory tract infection [7]. We hypothesize that reduced exposure to these viruses may have resulted in a change in clinical manifestations before and after pandemic onset.

Despite this difference in clinical manifestations, there were no differences in the age of infected participants or in the seasonal patterns of PIV circulation following pandemic onset. Non-pharmaceutical interventions put in place to reduce risk of transmission of SARS-CoV-2 led to a reduction in circulation of other respiratory pathogens, such as RSV, with a resurgence of infection (particularly in children older than 1 year of age) seen when restrictions were lifted [12,13]. Data regarding changes in PIV circulation as a result of the use of non-pharmacologic interventions show mixed results. Li et al. and Cho et al. both describe a disappearance of PIV circulation with pandemic onset, followed by a large out of season PIV epidemic shortly after relaxation of non-pharmaceutical interventions [12,13]. Contrary to this finding, and more consistent with our data, Zhao reports that while seasonal prevalence patterns of most respiratory viruses changed with the lifting of the SARS-CoV-2 restriction measures, PIV epidemiology was not substantially impacted by these changes [14]. These observed differences in epidemiology highlight the importance of monitoring regional virus circulation and avoiding extrapolation or overgeneralization of distinct, albeit similar, reported epidemiological patterns.

We found that PIV detection was negatively associated with rainfall, and PIV 3, specifically, was detected with higher frequency during the dry months compared to the rainy months. This is contrary to previous reports that show either no correlation or a positive association between PIV detection and precipitation [15,16,17,18]. It is important to note that the dry months with the highest frequency of PIV cases also were there the cooler months of the year. PIV detection was also associated with lower ambient temperature, a finding that held true through both univariate and multivariate logistic regression analyses, suggesting that temperature was among the main factors influencing PIV circulation and detection. While most of the prior reports show either no association or a positive association between PIV detection and mean temperature, Ciu et al. did find that PIV detection was negatively associated with average temperatures [19]. It is difficult to identify reasons for the varying meteorologic associations with PIV infection across the globe given the scarcity of available data.

Limitations to this study include a sample size that may not be large enough to detect differences in meteorologic factors associated with PIV circulation. Further, this study was conducted at a single site in a city with a climate that differs from the remaining regions of the country, as the coastal areas tend to experience a tropical climate, the nearby mountains have a temperate climate, while the Amazon region has a rainforest climate. The differences in daily temperatures, humidity, and rainfall can impact PIV epidemiology across the country. These factors limit generalizability of the data to other regions, however, are important to report to understand PIV epidemiology in this community. In this study, we included subjects whose nasopharyngeal samples co-detected other pathogens in addition to PIV. We also did not collect information related to past medical history or perinatal complications and therefore are unable to account for potential confounding factors, such as a history of asthma, low birth weight, or prematurity.

## 5. Conclusions

Parainfluenza viruses contribute to the disease burden of medically attended acute respiratory infections, including bronchiolitis, among young children in Machala, Ecuador. PIV 3, the most frequent of the virus types detected, was most prevalent during months with the coolest temperatures and the least rainfall. These findings can contribute to public health guidance regarding implementation of disease prevention measures, improving evaluation of healthcare delivery resources, and promote judicious use of antibiotics for acute outpatient respiratory tract illnesses.

## Figures and Tables

**Table 1 ijerph-22-00821-t001:** Demographic and clinical illness characteristics of study participants.

	Total	PIV * Detected	PIV Not Detected	*p* **
Study participants	1251	117	1134	
Mean age in months (range)	19, 1–59	18, 1–59	19, 1–59	0.32
Male gender, n (%)	695 (56)	66 (56)	629 (55)	0.85
Symptoms				
Fever, n (%)	1017 (81)	95 (81)	922 (81)	0.98
Nasal congestion, n (%)	1186 (95)	110 (94)	1076 (95)	0.69
Cough, n (%)	641 (51)	70 (60)	570 (50)	0.05
Wheeze, n (%)	93 (7)	20 (17)	73 (6)	<0.0001
Mean symptom duration in days (range)	3.3, 1–10	3.3, 1–7	3.3, 1–10	0.8
Upper respiratory tract infection, n (%)	1081 (86)	96 (82)	956 (84)	0.17
Nasopharyngitis, n (%)	1009 (81)	92 (79)	931 (82)	0.36
Laryngotracheitis, n (%)	4 (<1)	4 (3)	25 (2)	0.34
Lower respiratory tract infection, n (%)	124 (10)	16 (14)	107 (9)	0.17
Bronchiolitis, n (%)	100 (8)	13 (11)	87 (8)	0.19
Bronchitis, n (%)	8 (1)	2 (2)	6 (1)	0.17
Pneumonia, n (%)	16 (1)	1 (1)	14 (1)	1
Received antibiotics, n (%)	176 (14)	10 (9)	165 (15)	0.75
Influenza vaccination, n (%)	646 (52)	63 (54)	583 (51)	0.62
Pertussis vaccination, n (%)	1104 (88)	106 (91)	998 (88)	0.41

* PIV parainfluenza virus; ** *p* represents the statistical difference between subjects whose nasopharyngeal sample tested positive for PIV and those who samples tested negative for PIV. *T*-test was performed to compare ages and symptom duration. Chi-square was performed when comparing categorical data.

**Table 2 ijerph-22-00821-t002:** Demographic and clinical illness characteristics of the 117 study participants testing positive for parainfluenza virus (PIV).

	PIV 1	PIV 2	PIV 3	PIV 4	*p* *
Study participants, n (%)	14 (12)	8 (7)	71 (61)	25 (21)	
Mean age in months (range)	23, 1–59	20, 3–59	18, 1–59	13, 1–48	0.22
Male gender, n (%)	8 (57)	6 (75)	39 (55)	14 (56)	0.76
Symptoms					
Fever, n (%)	9 (64)	8 (100)	61 (86)	18 (72)	
Nasal congestion, n (%)	12 (86)	8 (100)	68 (96)	23 (92)	
Cough, n (%)	8 (57)	5 (63)	40 (56)	18 (72)	0.58
Wheeze, n (%)	6 (43)	1 (13)	6 (8)	7 (28)	<0.01
Mean symptom duration in days (range)	3.5, 2–6	2.9, 2–7	3.1, 1–7	3.8, 1–7	0.19
Upper respiratory tract infection, n (%)	7 (50)	7 (88)	64 (90)	18 (72)	0.02
Nasopharyngitis, n (%)	7 (50)	7 (88)	63 (89)	16 (60)	<0.001
Laryngotracheitis, n (%)	0	0	1 (1)	3 (12)	
Lower respiratory tract infection, n (%)	4 (29)	1 (13)	5 (7)	7 (28)	0.02
Bronchiolitis, n (%)	3 (21)	0	4 (6)	6 (24)	
Bronchitis, n (%)	1 (7)	1 (13)	0	1 (4)	
Pneumonia, n (%)	0	0	1 (1)	0	
Received antibiotics, n (%)	1 (7)	1 (13)	8 (11)	0	
Influenza vaccination, n (%)	10 (71)	4 (50)	40 (56)	10 (40)	0.27
Pertussis vaccination, n (%)	13(93)	8 (100)	66 (93)	20 (80)	

* *p* represents the statistical difference by parainfluenza virus type. One way ANOVA testing was performed to test differences in means. Chi-square testing was performed to compare categorical data.

**Table 3 ijerph-22-00821-t003:** Demographic and clinical illness characteristics of enrolled subjects testing positive for detection of parainfluenza virus (PIV) before and after the onset of the SARS-CoV-2 pandemic.

	PIV Detected			
	All Subjects Who Tested Positive for PIV Detection	Subjects Enrolled Prior to the Onset of the SARS-CoV-2 Pandemic	Subjects Enrolled After the Onset of the SARS-CoV-2 Pandemic	*p* *
Number of enrolled subjects	117	57	60	
Mean age in months, (range)	18, 1–60	16, 1–60	19, 1–60	0.22
Male gender, n (%)	66 (56)	32 (56)	34 (57)	0.95
Symptoms				
Fever, n (%)	95 (81)	37 (65)	58 (97)	<0.00001
Nasal congestion, n (%)	110 (94)	53 (93)	57 (95)	0.7
Cough, n (%)	70 (60)	45 (79)	25 (42)	<0.0001
Wheeze, n (%)	20 (17)	18 (32)	2 (3)	0
Mean symptom duration in days, range	3.3, 1–7	3.6, 1–7	3, 1–7	0.02
Upper respiratory infection	96 (82)	37 (65)	59 (98)	<0.001
Nasopharyngitis, n (%)	92 (79)	34 (60)	58 (97)	<0.00001
Laryngotracheitis, n (%)	4 (3)	3 (5)	1 (2)	0.4
Lower respiratory infection	16 (14)	14 (25)	2 (3)	<0.001
Bronchiolitis, n (%)	13 (11)	11 (19)	2 (3)	<0.01
Bronchitis, n (%)	2 (2)	2 (4)	0	0.24
Pneumonia, n (%)	1 (1)	1 (2)	0	0.5
Received antibiotics, n (%)	10 (9)	4 (7)	6 (10)	0.7
Influenza vaccination, n (%)	63 (54)	33 (58)	30 (50)	0.4
Pertussis vaccination, n (%)	106 (91)	53 (93)	53 (88)	0.53

* *p* represents the statistical difference between subjects with PIV detected before the onset of the SARS-CoV-2 pandemic and those with PIV detected after the pandemic onset. Chi-square was performed to compare the categorical data. *T*-test was performed to compare age and symptom duration.

**Table 4 ijerph-22-00821-t004:** Meteorologic factors associated with parainfluenza virus (PIV) detection.

Variable	PIV Detected	PIV Not Detected	AUC *	*p* **
Mean temperature				
n	72	137	0.642	0.001
Mean (SD)	25.1 (2.1)	26.2 (2.1)		
Median (IQR)	24.6 (3.6)	26.8 (3.5)		
Min, Max	22.2, 29.1	22, 29.8		
Q1, Q3	23.4, 27	24.5, 28		
Mean dew point				
n	72	137	0.454	0.276
Mean (SD)	21.6 (1.5)	21.9 (1.6)		
Median (IQR)	22 (2.3)	21.9 (2.8)		
Min, Max	19, 24.7	18.8, 24.9		
Q1, Q3	20.2, 22.6	20.4, 23.2		
Mean relative humidity (%)				
n	64	120	0.499	0.979
Mean (SD)	77 (7.5)	77.1 (6.2)		
Median (IQR)	77.2 (12.7)	77.4 (9.1)		
Min, Max	61.8, 92.2	63.3, 88.6		
Q1, Q3	70.2, 82.9	72.8, 81.8		
Humidity Range (%)				
n	57	118	0.61	0.019
Mean (SD)	35.8 (30.4)	24.5 (22.2)		
Median (IQR)	21.7 (48.6)	13.1 (36.7)		
Min, Max	4.1, 91.7	3.6, 91.7		
Q1, Q3	7.9, 56.6	6.2, 42.8		
Mean barometric pressure				
n	72	137	0.612	0.008
Mean (SD)	1011.4 (1.2)	1010.8 (1.3)		
Median (IQR)	1011.5 (1.9)	1010.9 (1.8)		
Min, Max	1008.7, 1013.5	1007.6, 1013.6		
Q1, Q3	1010.3, 1012.3	1009.9, 1011.7		
Min, Max	0, 0.3	0, 0.2		
Q1, Q3	0, 0.07	0, 0.04		

* AUC is the area under the Receiver Operating Characteristics (ROC) curve to classify the dissimilatory capacity of the variable to assess whether PIV was detected. ** *p* values to test whether the AUC is different from 0.5, the AUC of a non-informative classifier. The test is equivalent to the Wilcoxon rank sum test.

**Table 5 ijerph-22-00821-t005:** Odds ratios of parainfluenza virus (PIV) detection events associated with meteorologic factors using univariate and multivariate logistic regression.

	Univariate Logistic Regression			Multivariate Logistic Regression		
	Odds Ratio	95% CI	*p*	Odds Ratio	95% CI	*p*
Mean temperature (C)	7.82 × 10^−1^	0.678, 0.898	0.001	6.32 × 10^−1^	0.473, 0.826	0.001
Humidity range (%)	1.017	1.005, 10.3	0.007	1.051	1.025, 1.082	0

## Data Availability

Dataset available on request from the authors.

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
