# Peer review of "Medically Attended Outpatient Parainfluenza Virus Infections in Young Children from a Single Site in Machala, Ecuador"

_ijerph, 2025, doi:10.3390/ijerph22060821_

Round 1
Reviewer 1 Report
Comments and Suggestions for Authors<Major flaws>
(1) When evaluating cases prone to wheezing, it would be methodologically rigorous to assess whether analyses accounted for potential confounding factors such as asthma history and perinatal complications (e.g., low birth weight/prematurity). While reviewing the Subjects and Methods section, we confirmed no explicit description of these adjustments either in the method section or in the authors’ previous publications.
If the dataset contains sufficient statistical power, incorporating these variables into multivariate analyses could strengthen the validity of findings. If inadequate sample sizes preclude such adjustments, explicitly acknowledging this limitation in the study’s Limitations section would enhance methodological transparency. This approach aligns with best practices for addressing potential biases in observational respiratory research.
(2) The report lacks explicit documentation on whether all included cases represent exclusive PIV detection or encompass instances of viral co-detection (e.g., multiple pathogens identified per patient). If the analysis incorporates cases with concurrent viral detection (particularly clinically impactful agents like RSV), this methodological approach may introduce confounding effects. To enhance interpretative validity, I propose a supplementary analysis focusing exclusively on PIV-mono-detected cases. When these data are limited, explicit acknowledgment of this constraint within the "Study Limitations" section would strengthen methodological transparency.
(3) It is hypothesized that cases enrolled after the COVID-19 pandemic may have had reduced exposure to parainfluenza viruses (PIVs) - including maternal exposure - due to global isolation measures implemented during the early pandemic period. This temporal reduction in PIV exposure could have contributed to differences in clinical manifestations between the pre- and post-pandemic eras. If previously reported, relevant studies should be cited and discussed.
(4)If available, authors are recommended to provide data on whether the proportion of PIV types changed before and after the COVID-19 pandemic. If a shift in type distribution is confirmed, including a discussion on whether this change influenced differences in clinical presentations between the pre- and post-pandemic periods would strengthen the manuscript's discussion section.
(5) The conclusion described in Line 258-262 are the complete opposite from the results in Line 176-182.
<Minor flaw>
Please refine the Figure 1 as it is incomplete.
Author Response
Comment 1: When evaluating cases prone to wheezing, it would be methodologically rigorous to assess whether analyses accounted for potential confounding factors such as asthma history and perinatal complications (e.g., low birth weight/prematurity). While reviewing the Subjects and Methods section, we confirmed no explicit description of these adjustments either in the method section or in the authors’ previous publications.
If the dataset contains sufficient statistical power, incorporating these variables into multivariate analyses could strengthen the validity of findings. If inadequate sample sizes preclude such adjustments, explicitly acknowledging this limitation in the study’s Limitations section would enhance methodological transparency. This approach aligns with best practices for addressing potential biases in observational respiratory research.
Response 1: We did not collect information related to past medical history, including asthma diagnoses or perinatal complications. The limitations were revised to include the following statement to address this, “In addition, we did not collect information related to past medical history or perinatal complications and therefore are unable to account for potential confounding factors, such as a history of asthma or low birth weight or prematurity.” These changes have been made on page 9-10, lines 303-305.
Comment 2: The report lacks explicit documentation on whether all included cases represent exclusive PIV detection or encompass instances of viral co-detection (e.g., multiple pathogens identified per patient). If the analysis incorporates cases with concurrent viral detection (particularly clinically impactful agents like RSV), this methodological approach may introduce confounding effects. To enhance interpretative validity, I propose a supplementary analysis focusing exclusively on PIV-mono-detected cases. When these data are limited, explicit acknowledgment of this constraint within the "Study Limitations" section would strengthen methodological transparency.
Response 2: To address the reviewer’s suggestion, the following paragraph has been added to the results section: “Of the 117 parainfluenza virus positive samples, 29 (25%) also detected at least one other pathogen: 18 (62%) rhinovirus/enterovirus, 5 (17%) RSV, 3 (10%) influenza virus, and 1 (3%) each of human coronavirus, adenovirus, and atypical bacteria. While most of these samples detected 2 co-pathogens, 2 samples detected 3 pathogens (PIV 3 with rhinovirus/enterovirus and RSV; PIV 1 with human metapneumovirus and adenovirus). There were no significant differences between subjects with pathogen co-detection or just detection of a parainfluenza virus with regards to age (mean 17.2 months vs 17.7 months, p 0.9), fevers (24/29 (83%) vs 71/88 (81%), p 0.8), nasal congestion (28/29 (97%) vs 82/88 (93%), p 0.7), cough (19/29 (66%) vs 51/88 (58%), p 0.5), wheeze (8/29 (28%) vs 12/88 (14%), p 0.08), or symptom duration (mean 3.6 days vs 3.2 days, p 0.3).” These changes can be found on page 6, lines 162-170.
(3) It is hypothesized that cases enrolled after the COVID-19 pandemic may have had reduced exposure to parainfluenza viruses (PIVs) - including maternal exposure - due to global isolation measures implemented during the early pandemic period. This temporal reduction in PIV exposure could have contributed to differences in clinical manifestations between the pre- and post-pandemic eras. If previously reported, relevant studies should be cited and discussed.
To address the reviewer’s suggestion, the following has been added to the discussion section: “Similarly, we previously reported that in the parent study, we found that participants whose nasopharyngeal samples detected RSV prior to the onset of the pandemic were more likely to report cough or wheeze and be diagnosed with a lower respiratory tract infection, while those infected with RSV after pandemic onset were more likely to report fever and be diagnosed with an upper respiratory tract infection [7]. We hypothesize that reduced exposure to these viruses may have resulted in a change in clinical manifestations before and after pandemic onset. “ These changes can be found on page 10, lines 264-269.
(4)If available, authors are recommended to provide data on whether the proportion of PIV types changed before and after the COVID-19 pandemic. If a shift in type distribution is confirmed, including a discussion on whether this change influenced differences in clinical presentations between the pre- and post-pandemic periods would strengthen the manuscript's discussion section.
As reported in the results section: “There were no significant changes in general or type-specific PIV seasonality following pandemic onset.” No shift in type distribution was identified. This can be found on page 7, lines 191-192.
(5) The conclusion described in Line 258-262 are the complete opposite from the results in Line 176-182.
This sentence has been corrected to read as follows: “Individuals with nasopharyngeal samples positive for the detection of PIV who were enrolled prior to the onset of the SARS-CoV-2 pandemic were more likely to report cough and more likely to be diagnosed with a lower respiratory tract infection, while those enrolled after the pandemic onset were more likely to report fever and more likely to be diagnosed with an upper respiratory tract infection.” These changes can be found on page 10, lines 261-264.
<Minor flaw>
Please refine the Figure 1 as it is incomplete.
Figure 1 has been uploaded separately given the formatting difficulties when trying to insert into the template.
Reviewer 2 Report
Comments and Suggestions for Authors
Overall, the manuscript is well-structured and the information is clearly presented. However, a few suggestions could enhance its clarity and impact: some sections of the Discussion are repetitive and may be difficult to follow; the study limitations should be further elaborated, particularly considering the small cohort size; and the Conclusions would benefit from a more comprehensive synthesis of the main findings and implications.
Author Response
Comment: Overall, the manuscript is well-structured and the information is clearly presented. However, a few suggestions could enhance its clarity and impact: some sections of the Discussion are repetitive and may be difficult to follow; the study limitations should be further elaborated, particularly considering the small cohort size; and the Conclusions would benefit from a more comprehensive synthesis of the main findings and implications.
Response:
The discussion has been revised to remove statements that are repetitive.
To address the reviewer’s suggestions, we have expanded the study limitations as follows: “Limitations to this study include a sample size that may not be large enough to detect differences in meteorologic factors associated with PIV circulation. Further, this study was conducted at a single site in a city with a climate that differs from the remaining regions of the country, as the coastal areas tend to experience a tropical climate, the nearby mountains have a temperate climate, while the Amazon region has a rainforest climate. The differences in daily temperatures, humidity, and rainfall can impact PIV epidemiology across the country. These factors limit generalizability of the data to other regions, however, are important to report to understand PIV epidemiology in this community. In addition, we did not collect information related to past medical history or perinatal complications and therefore are unable to account for potential confounding factors, such as a history of asthma, low birth weight, or prematurity.” These changes can be found on pages 10-11, lines 297-305.
We have revised the conclusion as follows: “Parainfluenza viruses contribute to the disease burden of medically attended acute respiratory infections, including bronchiolitis, among young children in Machala, Ecuador. PIV 3, the most frequent of the virus types detected, was most prevalent during months with the coolest temperatures and the least rainfall. These findings can contribute to public health guidance regarding implementation of disease prevention measures, improving evaluation of healthcare delivery resources, and promote judicious use of antibiotics for acute outpatient respiratory tract illnesses.” These changes can be found on page 11, lines 307-312.
Reviewer 3 Report
Comments and Suggestions for Authors
This study contributes valuable data to the existing body of knowledge on pediatric PIV infections. The reviewers make the following recommendations to provide context and clarity to the manuscript:
- The single-site nature of the study significantly limits the generalizability of the data and should be more explicitly emphasized throughout the sections of the paper, as well as be considered for inclusion in the title of the manuscript
- The reported 9% incidence of PIV in the study population seems contrary to the claims made regarding substantial contribution of PIV to morbidity (Line 26). Consider providing additional location-specific context for cost or comparison to local incidence rates and other published epidemiological data
- Consider reframing the discussion around the effects of the pandemic. There is no information provided regarding whether patients were co-infected, therefore presenting the demographic and clinical characteristic breakdown into "before" and "after" the pandemic is of unclear relevance. If there were hypotheses regarding the impact of the pandemic on seasonality of PIV infections, inclusion of time frame in the discussion may help to make this point
Line 46-50: This summarizes the purpose of the study well. Reiterating this framing in the conclusion may help to reinforce the relevance of the findings and provide a cohesive ending.
- Figure 1: This figure requires reformatting to improve readability and interpretation. Recommend inclusion of a legend to explain the color scheme as well as adding tick labels along the axes to better convey any trends.
- Line 295: Consider adding brief discussion on how the climate differs from the remaining regions of the country, and how that impacts the climate-trends related to PIV
English is fine.
Author Response
This study contributes valuable data to the existing body of knowledge on pediatric PIV infections. The reviewers make the following recommendations to provide context and clarity to the manuscript:
Comment 1: The single-site nature of the study significantly limits the generalizability of the data and should be more explicitly emphasized throughout the sections of the paper, as well as be considered for inclusion in the title of the manuscript
Response 1: We have modified the title to read as follows: Medically attended outpatient parainfluenza virus infections in young children from a single site in Machala, Ecuador. We have added the word “single” to qualify the single study site in the abstract (page 1, line 17), methods (page 2 line 61), and limitations (page 10, lines 298-299) section of the manuscript.
The reported 9% incidence of PIV in the study population seems contrary to the claims made regarding substantial contribution of PIV to morbidity (Line 26).
We have revised the sentence in Line 26 as follows: “Our findings highlight the contribution that PIVs play in the morbidity associated with pediatric medically attended outpatient respiratory tract infection...”
Consider providing additional location-specific context for cost or comparison to local incidence rates and other published epidemiological data.
These data were added to the discussion as follows: “We detected PIV in 9% of subjects seeking medical attention for an acute respiratory infection, a rate higher than what has been previously reported [8, 9]. Azziz-Baumgartner found that 6% of children ages 1-4 years with a medically attended acute respiratory tract infection in northwest Ecuador tested positive for parainfluenza virus [8]. A respiratory pathogen surveillance study found that 1.5% of tested samples were positive for parainfluenza virus in Ecuador’s coastal region [9]. Differences in diagnostic testing likely account for the increase in PIV detection in our study with the use of multiplex-PCR assays, which are more sensitive for virus detection than the direct immunofluorescence assays used in the prior studies. Improved diagnostic testing methods lead to a truer understanding of virus-specific epidemiology, data which can be used to guide both prevention and treatment measures. One study published in 2017 found that 57/406 (14%) of children under 5 years of age admitted to the hospital in Quito, Ecuador, for severe pneumonia tested positive for a parainfluenza virus [10]. It is important to note, however, the difference in the study population as Jonnalagadda’s study focused on hospitalized children and our study on children seeking ambulatory medical care.” These changes can be found on page 9, lines 241-253.
Consider reframing the discussion around the effects of the pandemic. There is no information provided regarding whether patients were co-infected, therefore presenting the demographic and clinical characteristic breakdown into "before" and "after" the pandemic is of unclear relevance. If there were hypotheses regarding the impact of the pandemic on seasonality of PIV infections, inclusion of time frame in the discussion may help to make this point
Information regarding co-infections have been added to the results section: “Of the 117 parainfluenza virus positive samples, 29 (25%) also detected at least one other pathogen: 18 (62%) rhinovirus/enterovirus, 5 (17%) RSV, 3 (10%) influenza virus, and 1 (3%) each of human coronavirus, adenovirus, and atypical bacteria. While most of these samples detected 2 co-pathogens, 2 samples detected 3 pathogens (PIV 3 with rhinovirus/enterovirus and RSV; PIV 1 with human metapneumovirus and adenovirus). There were no significant differences between subjects with pathogen co-detection or just detection of a parainfluenza virus with regards to age (mean 17.2 months vs 17.7 months, p 0.9), fevers (24/29 (83%) vs 71/88 (81%), p 0.8), nasal congestion (28/29 (97%) vs 82/88 (93%), p 0.7), cough (19/29 (66%) vs 51/88 (58%), p 0.5), wheeze (8/29 (28%) vs 12/88 (14%), p 0.08), or symptom duration (mean 3.6 days vs 3.2 days, p 0.3).” These changes can be found on page 6, lines 162-170.
In addition, the hypotheses and interest in assessing for changes in PIV infection epidemiology before and after pandemic onset has been added to the discussion as follows: “ Similarly, we previously reported that in the parent study, we found that participants whose nasopharyngeal samples detected RSV prior to the onset of the pandemic were more likely to report cough or wheeze and be diagnosed with a lower respiratory tract infection, while those infected with RSV after pandemic onset were more likely to report fever and be diagnosed with an upper respiratory tract infection [7]. We hypothesize that reduced exposure to these viruses may have resulted in a change in clinical manifestations before and after pandemic onset. “ These changes can be found on page 10, lines 264-269.
Line 46-50: This summarizes the purpose of the study well. Reiterating this framing in the conclusion may help to reinforce the relevance of the findings and provide a cohesive ending.
To address the reviewer’s suggestion, the following statement has been added to the conclusion: “These findings can contribute to public health guidance regarding implementation of disease prevention measures, improving evaluation of healthcare delivery resources, and promote judicious use of antibiotics for acute outpatient respiratory tract illnesses.” These changes can be found on page 11, lines 307-312.
Figure 1: This figure requires reformatting to improve readability and interpretation. Recommend inclusion of a legend to explain the color scheme as well as adding tick labels along the axes to better convey any trends.
Figure 1 has been uploaded separately given the formatting difficulties when trying to insert into the template.
Line 295: Consider adding brief discussion on how the climate differs from the remaining regions of the country, and how that impacts the climate-trends related to PIV
To address the reviewer’s suggestions, the following has been added to the limitations section of the discussion: “Further, this study was conducted at a single site in a city with a climate that differs from the remaining regions of the country, as the coastal areas tend to experience a tropical climate, the nearby mountains have a temperate climate, while the Amazon region has a rainforest climate. The differences in daily temperatures, humidity, and rainfall can impact PIV epidemiology across the country.” These changes can be found on page 10, lines 298-302.
Round 2
Reviewer 1 Report
Comments and Suggestions for Authors
Authors' responses to the comment 1, 3, 4, and 5 are acceptable.
On the other hand, further revision is necessary regarding Comment 2. Authors performed additional analyses by separating cases into those with PIV single-positive and those with co-detection of other viruses. However, this was done only for data represented in Table 1. Ideally, similar analyses should also be conducted for the data presented in Tables 2 to 4-particularly for comparisons of clinical features, such as the presence or absence of wheezing before and after the pandemic, as shown in Table 3-by distinguishing between PIV single-positive cases and those with co-detection of other viruses. If it is feasible, authors should restrict the analysis to cases with PIV single positivity. If this is not possible, it is necessary to clearly state in the limitations of the study that cases with co-positivity for other viruses are also included.
Author Response
Comment 1: On the other hand, further revision is necessary regarding Comment 2. Authors performed additional analyses by separating cases into those with PIV single-positive and those with co-detection of other viruses. However, this was done only for data represented in Table 1. Ideally, similar analyses should also be conducted for the data presented in Tables 2 to 4-particularly for comparisons of clinical features, such as the presence or absence of wheezing before and after the pandemic, as shown in Table 3-by distinguishing between PIV single-positive cases and those with co-detection of other viruses. If it is feasible, authors should restrict the analysis to cases with PIV single positivity. If this is not possible, it is necessary to clearly state in the limitations of the study that cases with co-positivity for other viruses are also included.
Response 1: We have added the following sentence to the limitations section: In this study, we included subjects whose nasopharyngeal samples co-detected other pathogens in addition to PIV.